# Effect of Impregnation of Biodegradable Polyesters with Polyphenols from *Cistus linnaeus* and *Juglans regia Linnaeus* Walnut Green Husk

**DOI:** 10.3390/polym11040669

**Published:** 2019-04-11

**Authors:** Malgorzata Latos-Brozio, Anna Masek

**Affiliations:** Lodz University of Technology, Institute of Polymer and Dye Technology, ul. Stefanowskiego 12/16, 90-924 Lodz, Poland

**Keywords:** biodegradable polyesters, polylactide (PLA), polyhydroxyalkanoate (PHA), natural polyphenol, *Cistus linnaeus* (*Cistus* L.), *Juglans regia Linnaeus* (*Juglans regia* L.), impregnation

## Abstract

The publication describes a process combining the extraction of plant material and impregnation of biodegradable polymers (polylactide (PLA) and polyhydroxyalkanoate (PHA)). As raw plant materials for making extracts, *Cistus* and green walnut husk were selected due to their high content of active phytochemicals, including antioxidants. The extracts used to impregnate polymers contained valuable polyphenolic compounds, as confirmed by FTIR and UV–Vis spectroscopy. After impregnation, the polymer samples showed greater thermal stability, determined by the differential scanning calorimetry (DSC) method. In addition, despite the presence of natural antibacterial and antifungal substances in the extracts, the polyester samples remained biodegradable. The manuscript also describes the effect of UV aging on the change of surface free energy and the color of polymers. UV aging has been selected for testing due to the high susceptibility of plant compounds to this degrading factor. The combination of the extraction of plant material and polymer impregnation in one process proved to be an effective and functional method, as both the obtained plant extracts and impregnated polymers showed the expected properties.

## 1. Introduction

Polyphenols are phytochemical compounds that occur mainly in vegetables, fruits, coffee, tea, chocolates, legumes, cereals and beverages. These compounds are plant secondary metabolites and their main function is as antioxidants [1,2]. Natural polyphenols are also known for many very beneficial health-promoting properties, such as, anticancer, anti-inflammatory, cardioprotective, antiaging, antiseptic, etc. [3,4,5,6].

The literature describes the use of plant phenolic compounds as natural stabilizers in polymers. The most promising antioxidants for polymers are flavonoids, carotenoids, other natural phenols and phenolic polymers, including lignin. Flavonoids have proven to be much more effective stabilizers than the hindered phenols used in industry [7]. In addition, polyphenols of plant origin can be successfully used as dyes and indicators of the aging time of polymeric materials [8,9]. Natural antioxidants are also used as functional additives in active food packaging [10]. Particularly noteworthy is the combination of environmentally friendly, biodegradable polymers and plant polyphenols [11]. This combination allows the formation of a fully natural polymeric material, which does not pollute the environment.

Two polymers, polylactide (PLA) and polyhydroxyalkanoate (PHA) were used for the studies presented in the publication. Polylactide is a typical biodegradable polyester, which is produced from various natural sources such as tapioca, corn, sugar beets, etc. [12]. This polymer is considered a good substitute for standard petrochemical polymers such as polystyrene (PS) and has a wide range of applications, including in the packaging industry [13]. Other important biodegradable polymers that can potentially replace plastics produced from petrochemical sources are polyhydroxyalkanoates (polyoxoesters of hydroxyalkanoic acids). PHA’s polymers are synthesized by some bacteria as intracellular storage compounds and are used in food packaging and biomedical applications [14,15].

One of the methods of introducing plant polyphenols to polymers is the application of these natural compounds to the surface of the polymeric material in the impregnation process. This is not a very common method. The introduction of active substances to polymers in melting extrusion and the coating process is more popular [16]. The literature describes supercritical impregnation of both petrochemical and biodegradable polymers with substances of plant origin. However, the use of the supercritical fluid impregnation technique for polymers with natural substances is limited. For this type of impregnation, plant extracts obtained from supercritical extraction (such as the extract from thyme) or single plant compounds (as chemical reagents) are most commonly used [16,17,18,19,20,21,22,23,24,25,26]. Petrochemical polyethylene terephthalate/polypropylene (PET/PP) films were impregnated with olive leaf extract in order to provide the antioxidant properties of the material [17]. Biodegradable polylactide and its composite and nanocomposite were impregnated in supercritical conditions, with plant substances, thymol and cinnamaldehyde, to obtain materials with antibacterial properties [16,18,19,20]. As proposed in this manuscript, the combination of extraction and solvent-based impregnation in one process is a scientific novelty and has not yet been referred to in the literature. Ethanol extraction was used. Ethanol was not evaporated to concentrate the extract and the whole solution (with plant materials) was used in the next step of the combined process—impregnation.

In this manuscript, extracts of *Cistus linnaeus* (*Cistus* L.) and green husk of walnut *Juglans regia Linnaeus* (*Juglans regia* L.) were used for impregnation biodegradable polymers. The utilized plants are rich in active polyphenol compounds. Therefore, their addition should stabilize the polymers.

*Cistus* species are perennial, dicotyledonous flowering shrubs in pink or white depending on the species [27,28,29,30]. Phytochemical studies on various *Cistus* species revealed the presence of active substances, generally involved in many biological activities, essentially in the prevention of oxidative stress. The main biologically active components of the herb are polyphenolic compounds such as gallic acid, rutin, different flavonoid aglycones, and flavan-3-ols, as well as catechin, epicatechin, gallocatechin, gallocatechin-3-gallate and oligomeric procyanidin B1 and B3 [31,32,33,34,35]. Due to a composition rich in polyphenols, the *Cistus* extract has significant antioxidant and antimicrobial properties [36,37,38]. 

The second plant material used for impregnation of polymers was green husks of walnut *Juglans regia* L. The material was chosen, as well as *Cistus*, due to the large number of active polyphenols. Green walnut husks are a by-product of nut production and are rarely used. However, walnut husks, like fruit, leaves and liqueurs produced from green fruit, have significant antioxidant and antimicrobial activity [39,40,41,42,43,44]. The properties of green walnut hulls are associated with the presence of phenolic compounds in the plant material: chlorogenic acid, caffeic acid, ferulic acid, sinapic acid, gallic acid, ellagic acid, protocatechuic acid, syringic acid, vanillic acid, catechin, epicatechin, myricetin, and juglone. Walnut husks are an easily accessible waste source of active polyphenolic compounds with the potential to protect health and with antimicrobial activity [45]. 

The aim of the research presented in this manuscript is the solvent-based impregnation of biodegradable polymers with extracts of *Cistus* and walnut hulls. The proposed method combines two processes: extraction of plant material and impregnation of polymers. Moreover, this approach to polymer impregnation has not been described in the literature so far. The method allows the introduction of active phytochemicals directly from the plant material to the surface of polymers, which aims to improve the properties of materials, including stability.

## 2. Materials and Methods

### 2.1. Reagents

The objects of the study were two biodegradable polymers: polylactide (PLA) and polymer P(3,4HB) 2001 from the polyhydroxyalkanoate group of polymers (PHA). Polylactide (PLA) IngeoTM Biopolymer 4043D PLA was obtained from Nature WorksTM (Minnetonka, MN, USA) and had the following properties: *T*_g_ = 55–60 °C, *T*_m_ = 145–160 °C, and Melt Flow Index (MFI) = 6 g/10 min. Polymer PHA was produced by Simag Holdings Ltd. (Hong Kong, China) and had properties: P(3,4HB) containing 12 mol% 4-hydroxybutyrate, the average M_w_ was approximately 520 kDa, melt volume rate MVR (melt volume rate) = 15–20 g/10 min, (assay conditions: temperature 170 °C, nominal load 2.16 kg), and a density of 1.25 g/cm^3^.

Two plant materials were used for impregnation: *Cistus* and green husk of walnut. Dried *Cistus* herb was bought in the pharmacy as a herb for brewing (Manufacturer: Radix-Bis Sp. z o.o. Poland, country of origin: Turkey, 100% cut herb *Cistus* L.). Green walnut husks were obtained from a local farmer and were collected from a 16-year-old *Juglans regia* L. tree from cultivation in central Poland.

### 2.2. Methods of Impregnation of PLA and PHA Samples with Plant Materials

Polymer samples were prepared prior to impregnation. Dried (12 h, 50 °C) granulates of PLA and PHA were extruded using a laboratory extruder, Brabender. Strips with a thickness of 1.6–1.8 mm were obtained. The temperature of the working chamber of the extruder was 180 °C for PLA and 160 °C for PHA. The extruded strips were weighed and dried to constant weight at 70 °C.

To the sulfonation flasks (Figure 1, 4), 200 g of plant materials (comminuted) and 1 L of 96% ethanol (Avantor Performance Materials Poland S.A (POCH S.A.), Poland, pure) were added. The mixture was heated (50 °C) and stirred (800 rpm) for 1 h. Solutions (15 mL) were used for further testing. Afterwards, the polymer samples dried to a constant mass and were put in the flasks. Impregnation was carried out for 4 h at 50 °C. After 4 h, the flask and its contents were left for 24 h at room temperature. After this time, the samples were dried in a dryer (70 °C) to constant weight. The reference sample contained the PLA and PHA impregnated only with ethanol, without plant materials. The impregnation efficiency [%] was calculated from the changes in the mass of the samples.

### 2.3. Analysis of Plant Extracts

The composition of the plant material extracts for the presence of active polyphenols was determined using FTIR and UV–VIS spectroscopy. The Nicoled 670 (Thermo Fisher Scientific, Waltham, MA, USA) spectrophotometer was used for the measurements. Samples of plant materials and extracts were placed at the output of infrared beams. Oscillating spectra were obtained, the analysis of which allows the determination of the functional groups with which the radiation interacted. The UV–VIS spectra of the *Cistus* and walnut husk extract solutions were recorded from a mixture of 0.5 mL of each extract plus 1.5 mL of 96% ethanol. The mixture was scanned at 190–1100 nm using a UV-spectrophotometer (Evolution 220, Thermo Fisher Scientific, Waltham, MA, USA).

The antioxidant properties of *Cistus* and walnut husk extracts were determined by ABTS and DPPH methods. The methods consist of the reduction of free radicals 2,2’-azino-bis(3-ethylbenzothiazoline-6-sulphonic acid (ABTS) and 2,2-diphenyl-1-picrylhydrazyl (DPPH). A detailed description of the methods and the authors are presented in other publications [46,47]. ABTS or DPPH radical (A%) scavenging ability was determined using the following Equation (1):
Inhibition (A%) = [((A_0_ − A_1_)/A_0_) × 100](1)
where A_0_ is the absorbance of the control, and A_1_ is the absorbance in the presence of the sample antioxidant.

The ability of *Cistus* and walnut husk extracts to reduce the ferric ion (Fe^3+^–TPTZ complex) under acidic conditions was determined using the ferric-reducing antioxidant power (FRAP) assay. Cupric ion-reducing antioxidant capacity was determined by a CUPRAC assay. The methods involve the reduction of transition metal ions:FRAP: Fe^3+^→Fe^2+^
CUPRAC: Cu^2+^→Cu^1+^

The FRAP and CUPRAC methods are described fully in other publications [46,47]. The ferric and cupric ion-reducing power was calculated as (2):
ΔA = A_AR_ − A_0_(2)
where: A_0_—absorbance of the reagent test, A_AR_—absorbance after reaction.

Absorbance measurements in the ABTS, DPPH, FRAP, CUPRAC methods were performed using a UV-spectrophotometer (Evolution 220, Thermo Fisher Scientific, Waltham, MA, USA).

### 2.4. Analysis of PLA and PHA Samples

#### 2.4.1. Determination of the Influence of Mold Fungi on Polymers

The following species of mold fungus, *Aspergillus niger*, *Paecilomyces varioti*, *Chaetomium globosum*, *Trichoderma viride* and *Penicillium funiculosum*, were used for the study. Two methods (A and B) were used in studies of the influence of mold fungi on the studied polymer material. The tests were carried out according to the PN-EN ISO 846 2002 standard “Plastics. Evaluation of the action of microorganisms”.

*Method A* (material resistance test) is used to assess the natural immunity of the tested material, in the case where there is no other nutrient and if the tested material is a source of food for microorganisms. The test material was placed on a defective medium (without a carbon source). Subsequently, a suspension of microorganisms was evenly applied to the surface of the medium and material samples. The samples were incubated at 28 °C and 80% relative atmospheric humidity for 3 months. During the incubation, the microbial growth on the surface of the material was evaluated.

*Method B* (determination of the fungostatic effect and the effect of surface dirt on the resistance) is used in the case of expected surface contamination and is aimed at checking the fungicidal properties of the tested material and the effect of surface soil dirt on its resistance. The test material was placed on a nutrient medium. A suspension of microorganisms was then applied evenly. The samples were incubated at 28 °C and 80% relative atmospheric humidity for 3 months. After this time, the growth of microorganisms on the surface of the medium and samples was assessed and the appearance of growth inhibition zones around the samples was observed.

#### 2.4.2. Differential Scanning Calorimetry (DSC)

Using the Mettler Toledo DSC analyzer (TA 2920; TA Instruments, Greifensee, Switzerland), the temperature ranges of the sample phase changes were determined. The glass transition temperature (*T*_g_), crystallization temperature (*T*_cc_), melting temperature of the crystalline phase (*T*_m_) and oxidation temperature (*T*_o_) were tested. The heat (Δ*H*) accompanying the phase changes was also determined. The samples (5–6 μg, placed in 100-μL open, aluminium crucibles) were heated from 0 to 200 °C at a rate of 20 °C/min under an argon atmosphere. After 10 min at 200 °C, the samples were cooled to 0 °C. Then, the gas was switched from argon to air (flow rate 50 mL/min), and the samples were heated to 350 °C. Before the measurement, the apparatus was calibrated based on the following standards: temperature scale based on n-octane and indium, heat exchange according to an inducible heat of 28.45 J/mg.

#### 2.4.3. Surface Free Energy of PLA and PHA Samples

The surface free energy of the polyesters (γs) was determined on the basis of measured contact angles. The measurements of the contact angles were made for liquids with different polarities: distilled water, diiodomethane and ethylene glycol. The test was made using the OEC 15EC goniometer (DataPhysics Instruments GmbH, Filderstadt, Germany). The polar component, dispersive component and surface free energy were determined using software module SCA 20. Measurements were made for samples before ageing and after UV aging.

#### 2.4.4. Changes in Color after UV Irradiation

Color measurements were performed to determine the color change of the polyesters after UV aging. The measurements were carried out using a CM-3600d spectrophotometer (Konica Minolta Sensing, Inc., Osaka, Japan). The result of test is the color described in the CIE-Lab space and the determination of the color in a system of three coordinates: L, a and b, where L is the lightness parameter (maximum value of 100 represents a perfectly reflecting diffuser; minimum value of zero represents the color black), a is the axis of red–green and b is the axis of yellow–blue. The a and b axes have no specific numerical limits. The change in color, dE*_ab_, was calculated as follows (3):(3)$$\mathrm{dE}{*}_{\mathrm{ab}}=\sqrt{\left(\Delta a^{2})+(\Delta b^{2})+(\Delta L^{2}\right)}$$

The following color parameters were also calculated using coordinates a, b, L: whiteness index (4) (numerical indicator used to indicate the degree of whiteness), hue and chroma. In the CIE Lab color space, two of the axes are perceptually orthogonal to lightness. Hue (6, 7) can be calculated together with chroma (5), transforming coordinates a and b from rectangular form to polar form. Hue is the angular component of the polar representation, while chroma is the radial component.

Whiteness index:(4)$$W_{i}=100-\sqrt{{\left(100-L\right)}^{2}+a^{2}+b^{2}}$$

Chroma:(5)$$C{*}_{\mathrm{ab}}=\sqrt{a^{2}+b^{2}}$$

Hue:(6)$$h{*}_{\mathrm{ab}}=\mathrm{arctg}(\frac{b}{a})$$
where a > 0 and b > 0
(7)$$h{*}_{\mathrm{ab}}=180+\mathrm{arctg}(\frac{b}{a})$$
where a < 0 and b > 0.

Color changes are documented by taking photos with the Leica stereoscopic microscope. The software Optaview was used to analyze and process photos of samples. The pictures show samples at 43× magnification.

#### 2.4.5. UV Aging

The test was performed using a UV 2000 apparatus from Atlas. The measurement lasted 100 h and consisted of two alternately repeating segments with the following parameters: daily segment (radiation intensity = 0.7 W/m^2^, temperature 60 °C, and duration 8 h) and night segment (no UV radiation, temperature = 50 °C and duration 4 h).

## 3. Results and Discussion

The tests were started by confirming the presence of active phytochemicals in solutions prepared for impregnation, as well as determining the antioxidant activity and the ability to reduce transition metal ions of *Cistus* and walnut husk extracts. The functional groups of plant compounds in raw materials (Figure 2A,A1) and extracts (Figure 2B,B1) were identified by FTIR spectroscopy. It is obvious that functional groups correlating with cellulose, hemicellulose and lignin (3600–3000 cm^−1^—OH stretching; 2860–2970 cm^−1^—C–H stretching; 1470–1430 cm^−1^—O–CH_3_; 1000 cm^−1^—C–O–C) have been found on FTIR spectra of *Cistus* and walnut husk. The 1730–1700 cm^−1^ range corresponds to alkyl, aliphatic and aromatic compounds, whereas 900–700 cm^−1^—C–H stretching from aromatic hydrogen compounds [48,49,50]. The reaction with ethanol allows the extraction of valuable phenolic compounds from plant materials. A particularly important group of active compounds are flavonoids whose functional groups are present on the FTIR spectra of plant extracts. The ranges 1612–1598 cm^−1^ and 1488–1452 cm^−1^ are characteristic of the aromatic ring vibration. Other characteristic bands that come from the phenol group vibrations are C–OH deformation vibrations (1370–1308 cm^−1^) and C–OH stretching vibrations (1172–1112 cm^−1^) [51]. Functional groups correlating with ethanol, which is the basis of a solution for polymer impregnation, have also been found on FTIR spectra of plant extract (3390 cm^−1^ and 3360 cm^−1^—OH groups, 2980 cm^−1^—CH groups, 1090 cm^−1^ and 1050 cm^−1^—C–O groups) [52,53]. 

The UV–Vis spectroscopy (Figure 3) also confirms the presence of phenolic compounds in the solution used for the impregnation of polyesters. The UV–Vis spectra of flavones and related glycosides show two strong absorption peaks at 300–380 nm and 240–280 nm [54]. The maximum absorbance of flavonol was identified at 350 nm [55]. Also, the peaks of active phenolic acids have a maxima of absorbance in the range of about 290–350 nm, e.g., gallic acid, ferulic, *p*-coumaric acids and vanillic acid [56,57]. Thus, the peaks at 303, 302 and 365 nm on the UV–Vis spectra of *Cistus* and walnut husk extracts correspond to a mixture of plant phenolic acids, flavones and their glycosides. The low intensity peak at 408 nm in the spectrum of walnut husk extract is characteristic of juglone, which occurs in plants from group *Juglans regia* L. [58]. The peak at 490 nm can be associated with the presence of colored compounds, e.g., carotenoids or chlorophyll [55]. It was found that chlorophylls a and b show a maximum absorbance in the range of 400–500 and 600–700 nm [59], so the peak at 665 nm on the spectra of the *Cistus* extract corresponds to these compounds.

Table 1 summarizes the results of the determination of antioxidant activity (ABTS and DPPH methods) and the ability to reduce transition metal ions (FRAP and CUPRAC methods) of the extracts of *Cistus* and walnut husk. Active plant compounds, whose presence in solutions confirmed FTIR and UV–Vis spectroscopy, showed high ability to reduce the free radicals ABTS and DPPH. The antioxidant activity of the *Cistus* extract was higher than the walnut husk extract and was 60.71 ± 3.04% for the ABTS method and 57.94 ± 2.90% for the DPPH method. Both extracts showed good ability to reduce transition metal Fe^3+^ and Cu^2+^ ions, higher than for iron ions (FRAP method). The application of natural phenolic compounds to the surface of polyesters, by impregnation, should provide polymers greater stability and resistance to degradation. The high antioxidant activity of polyphenols should stop the unfavorable oxidation processes associated with the degradation of polymers. Transition metal ions can catalyze the aging reactions of polymers, so the ability of phenolic compounds to reduce iron and copper ions is a useful property in the stabilization of polymeric materials.

Figure 4 shows the impregnation efficiency of polyesters. Higher impregnation efficiency with both extracts (around 10%) was found for polyhydroxyalkanoate. The obtained impregnation efficiency for PLA and PHA are satisfactory and the number of phytochemicals impregnated on the surface of the polymers is sufficient to ensure stability for polymers. Compared to another work, one part by weight of antioxidants, added to the mass of polyesters during extrusion, gives them good resistance to oxidation [11]. Degradation of the polymer starts from its surface and then degrades deeper parts of the sample. Therefore, protecting the surface of the sample with antioxidants seems to be a better solution than introducing antioxidant compounds into the mass of the polymer. The obtained results may be related to the presence of a greater number of carbonyl groups in the PHA polymer, that may potentially combine with polyphenolic compounds, e.g., by oxygen bridges. Higher impregnation efficiency of PHA samples can also be caused by the more crystalline structure of the material. The polyhydroxyalkanoate is more crystalline in nature than the polylactide. PLA is an amorphous polymer and the characteristics of the polymeric material can affect the impregnation efficiency.

The crystallinity of polymeric materials can be determined by differential scanning calorimetry and knowledge of the melting enthalpy of 100% crystalline 100% polymer. The degree of crystallinity X_c_ can be calculated as follows: X_c_ (%) = ∆H_m_/∆H_0_ × 100, where ∆H_m_ is the melting enthalpy of the test polymer and ∆H_0_ is the melting enthalpy of 100% crystalline 100% PLA or PHA. The melting enthalpy estimated for 100% crystalline PLA is 97.3 J/g to 148 J/g [60]. The 148 J/g was used for calculations. The melting enthalpy of 100% crystalline polyhydroxybutyrate (PHB) is 146 J/g [61] and was used as the melting enthalpy of 100% crystalline PHA. The PHA polymer that used according to the manufacturer’s data is P(3,4HB). The data from the DSCs shown in the author’s publication [11] was used to calculate the degree of the crystallinity of PHA. The melting enthalpy of PHA was 40.4 J/g. The degrees of crystallinity are X_c_^PHA^ = 27% and X_c_^PLA^ = 17%. The results may be subject to error due to the assumed values of 100% crystallinity of both polymers.

For both PLA and PHA polyesters, a higher impregnation efficiency was observed with *Cistus* extract, which may be due to the greater amount of phenolic compounds present in this extract or the presence of compounds with better accessibility attaching more easily to polymers. The easier attachment of phenolic compounds to the polymers may be related to the more hydrophilic nature of some phytochemicals, facilitating reactions with hydrophilic polymers.

A relationship between antioxidant activity and impregnation efficiency was observed. The higher antioxidant activity of the *Cistus* extract correlates with the higher efficiency of polyester impregnation with extracts of *Cistus*. The higher antioxidant activity of *Cistus* and higher efficiency of the impregnation of *Cistus* samples may be the result of a larger amount of plant active compounds.

According to the literature, cited in the introduction of the manuscript, phytochemicals contained in *Cistus* and walnut husk can induce a microbicidal and fungicidal effect. Therefore, it was considered reasonable to analyze whether polymeric materials (on the example of PLA) had preserved their biodegradation properties after impregnation (Table 2). In the first stage (method A), a test variant was used in which no additional carbon source was added to the nutrient solution. The purpose of the experiment was to determine whether the test material is used as a nutrient by the fungus and how the compounds in the polymers interact with the microorganisms studied. The reference sample PLA was not a medium for microorganisms (growth intensity *GI* = 0 [a.u.]). The PLA/walnut husk sample contained medium substances or was contaminated to a small degree, allowing a slight increase of microorganisms (*GI* = 1 [a.u.]), while the PLA/*Cistus* sample was not resistant to microorganisms and contained substances that constituted a medium for their development (*GI* = 2 [a.u.]). The presence of plant materials on the surface of polylactide samples increased its susceptibility to biodegradation.

The method B of the research was aimed at determining whether mold fungi can grow on the tested materials when there is a carbon source (in the form of microbial culture). The conditions used in the experiment are similar to those occurring in nature, where dust particles settle on the surfaces, carrying with them carbon compounds. The intensity of mold fungi growth assessed as *GI* = 2 and 3 [a.u.] indicates the absence of a fungicidal effect of PLA samples. Good susceptibility of samples to biodegradation and no fungistatic effect may be caused by limited and low digestibility of phytochemicals responsible for microbicidal and fungicidal effects. The low persistence of these phytochemicals (including juglone) may be reflected in the high susceptibility of fresh, raw walnut husks to mold growth. According to the impregnation methodology described in the manuscript, ethanol extraction of plant material was performed in the first step, followed by impregnation. Before impregnation, the plant material was not removed from the flasks to increase the impregnation efficiency. Small fragments of plant materials could probably have been attached to the polyester samples. Plant materials are a source of carbon, which could have contributed to the growth of microorganisms. Small pieces of plants could be filtered out of the solution before impregnation. However, the assumption of the combination of the process was the impregnation of polymers with phytochemicals directly from plant material. Leaving the *Cistus* and walnut husk in the impregnation process was to increase the amount of active substances in the solutions and direct adhesion of phytochemicals from plants to polymers with a carrier EtOH.

The next step was the analysis of the stability of polymeric materials, determined in the example of polylactide samples. Differential scanning calorimetry (DSC) results are shown in Table 3. The impregnation of PLA with extracts of *Cistus* and walnut husk did not significantly affect the glass transition temperature (*T*_g_), crystallization temperature (*T*_cc_) and melting temperature (*T*_m_). However, the addition of plant compounds clearly increased the initial oxidation temperatures (*T*_o_)—approximately 16 °C for the PLA/*Cistus* sample and approximately 20 °C for the PLA/walnut husk. Higher oxidation temperatures should improve the oxidation resistance of the samples and increase the stability of the materials. What is more, higher oxidation temperatures testify to the presence of natural plant antioxidants on the surface of the samples.

Phytochemicals, especially flavonoids, are sensitive to UV irradiation, which can cause their polymerization or degradation. These processes are accompanied by a change in the properties of plant compounds. For this reason, the properties of impregnated polymers (change of surface free energy and color change) after controlled UV aging were determined. Figure 5 shows changes in the polar and dispersive components and surface free energy of PLA and PHA samples after UV irradiation. The surface free energy of polymeric materials is the sum of two components: polar and dispersion, and was determined by the Owens, Wendt, Rabel and Kälble (OWRK) method based on the determination of contact angles using test liquids with different polarity. The surface free energy of polyesters impregnated with *Cistus* and walnut husk extracts was reduced, compared to PLA and PHA reference samples (PLA 87 mJ/m^2^, PLA/*Cistus* 79 mJ/m^2^, PLA/walnut husk 57 mJ/m^2^; PHA 81 mJ/m^2^, PHA/*Cistus* 80 mJ/m^2^, PHA/walnut husk 56 mJ/m^2^). Introduction of active plant compounds to the surface of polymers may increase the number of polar groups. Unexpectedly, the surface free energy of samples after UV aging decreased, which may be due to different coefficients of thermal expansion of samples, as well as a heterogeneous arrangement of test samples. The surface of the samples was not covered completely with plant extracts. The uniformity of the impregnated samples was evaluated on the basis of visual observations and optical microscopy (Figure 6). During the observation, the surface of the samples covered with plant extracts was measured and referred to the entire surface of the samples. The degree of coverage of samples with plant extracts was estimated at 90%.

Color change is an important parameter, being a visual sign of changes in other properties of polymers, including those caused by degradation factors. Figure 7 shows changes in the color parameters of polyesters after UV aging. During UV irradiation, the temperature was increased to 60 °C. The color change of samples results from the combination of two factors, UV radiation and higher temperature. However, the main factor influencing the color change of the samples is UV radiation. The larger changes in color change coefficient, dE*_ab_, of samples impregnated with extracts from plant materials are obviously caused by the greater susceptibility of plant compounds to the color change under the influence of UV radiation. Phytochemicals can change color under the influence of UV irradiation, which is a common phenomenon in the natural environment. Furthermore, UV aging caused significant changes in the whiteness index and hue parameters of the samples. The visualization of the color change of the PLA and PHA samples is shown in Figure 6, as pictures taken with the optical microscope at 43× magnification. The photographs show samples before (left part) and after (right part) UV aging. Impregnation of polyesters with plant extracts caused a slightly green color of materials, especially samples impregnated with walnut husk extract. The particularly high coloring of the polyester with walnut husk extract was expected because the juglone found in various parts of the walnut tree is known as C.I Natural Brown 7 dye, used in hair dyeing in the cosmetics industry as well as in dyeing fibers.

## 4. Conclusions

The method presented in the manuscript combines the extraction of plant material and the impregnation of polymers in one process and has not been described in the literature so far. The process is effective and functional because both obtained extracts and impregnated polymeric materials have the expected properties. Plant extracts from *Cistus* and walnut husk were rich in polyphenols, as demonstrated by FTIR and UV–Vis spectroscopy. Due to the presence of active phytochemicals, the extracts were characterized by very good antioxidant activity (ABTS and DPPH methods), as well as the ability to reduce the ions transition metals, iron (FRAP) and copper (CUPRAC). Polymer impregnation with plant extracts resulted in materials with higher thermal stability. Moreover, despite the presence of microbicidal and fungicidal substances in the extracts used for impregnation, the polyester samples remained biodegradable. A thin film of plant extracts applied on polymers, probably also containing small fragments of plants, even increased the susceptibility of samples to biodegradation. In addition, this impregnation allows the formation of materials with a green color. The process may be a method of using plant by-products, such as walnut husk.

The combination of extraction and impregnation in one process allows the introduction of active phytochemicals directly from plant material to polymers. Furthermore, the combined process of extraction and impregnation is technically much easier and cheaper than the separate extraction of plant compounds and then introducing these substances to polymers by impregnation. The combined process is performed in one apparatus and does not require special equipment and conditions. From an industrial point of view, the introduction of phytochemicals in polymers in a combined extraction and impregnation process is a much easier and more economically advantageous solution than separate processes.

## Figures and Tables

**Figure 1 polymers-11-00669-f001:**
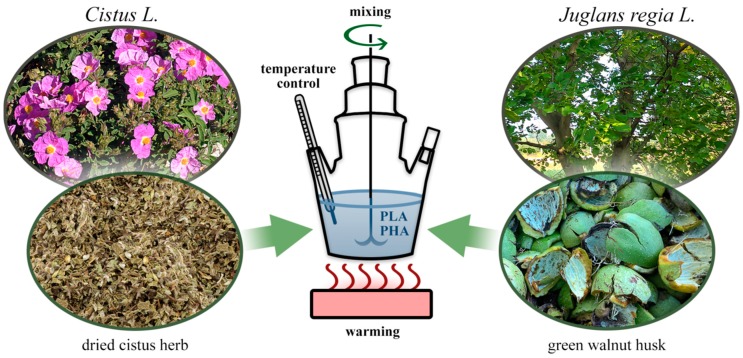
Scheme of apparatus and plant materials used in the impregnation of polymers.

**Figure 2 polymers-11-00669-f002:**
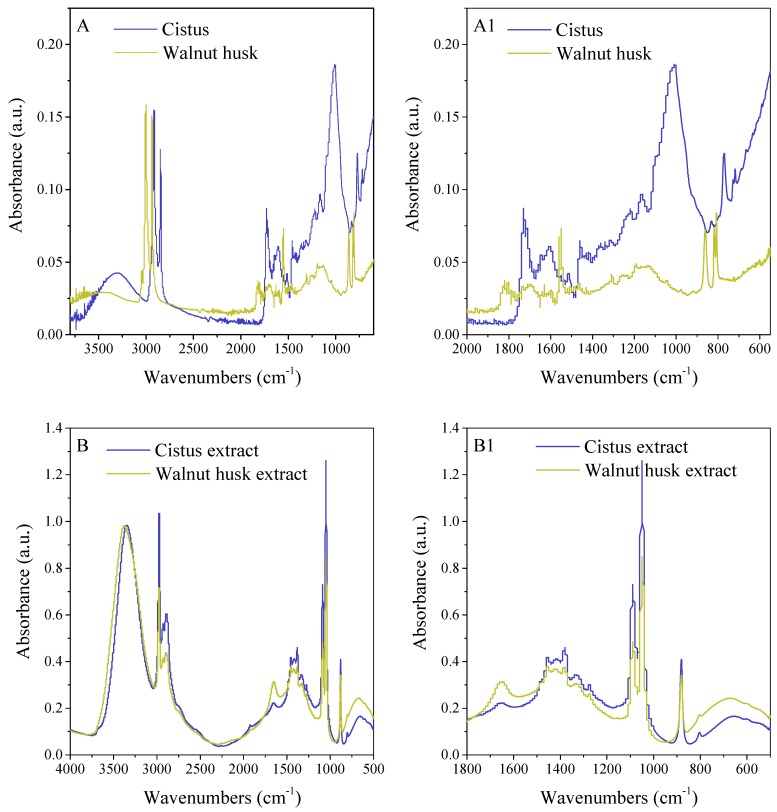
The FTIR spectra of plant materials (**A**,**A1**) and extracts of *Cistus* and walnut husk (**B**,**B1**).

**Figure 3 polymers-11-00669-f003:**
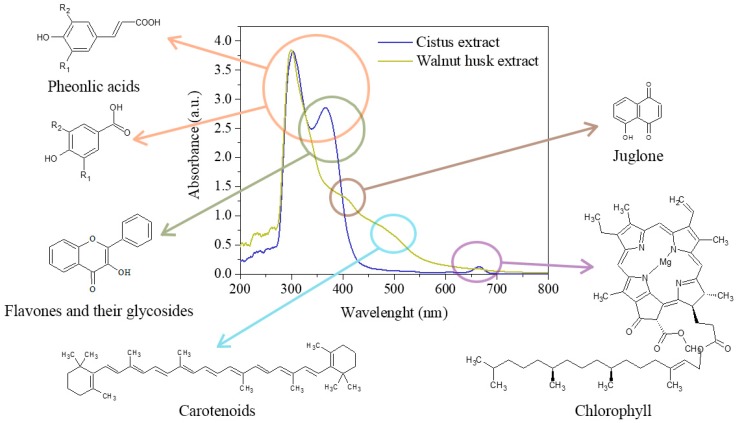
The UV–Vis spectra of the extracts of *Cistus* and walnut husk.

**Figure 4 polymers-11-00669-f004:**
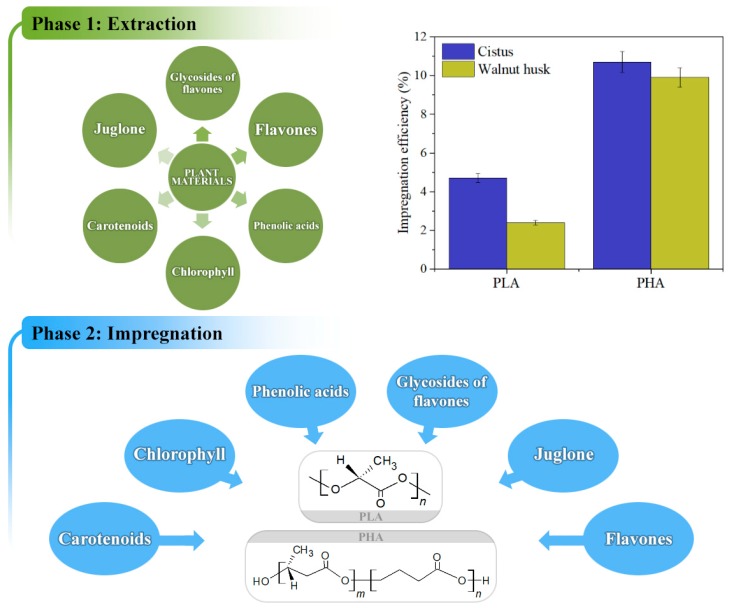
Impregnation efficiency of polyesters with extracts of *Cistus* and walnut husk. Scheme of the combined process of extraction of plant materials and impregnation of polymers—Phase 1: extraction of phytochemicals from *Cistus* and walnut husk, Phase 2: impregnation of PLA and PHA with plant compounds from extracts.

**Figure 5 polymers-11-00669-f005:**
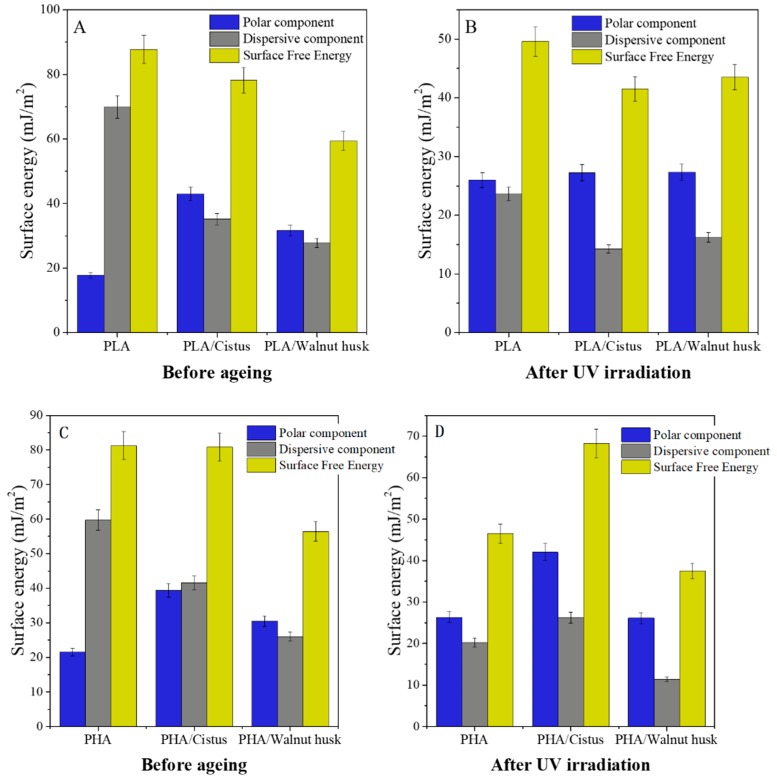
Surface free energy of PLA samples before (**A**) and after (**B**) UV irradiation. Surface free energy of PHA samples before (**C**) and after (**D**) UV irradiation.

**Figure 6 polymers-11-00669-f006:**
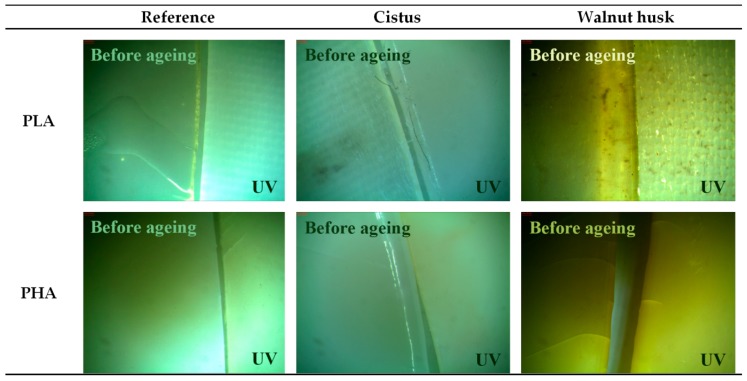
Visualization of the color change of PLA and PHA samples before and after UV aging. Photographs taken with an optical microscope at 43× magnification.

**Figure 7 polymers-11-00669-f007:**
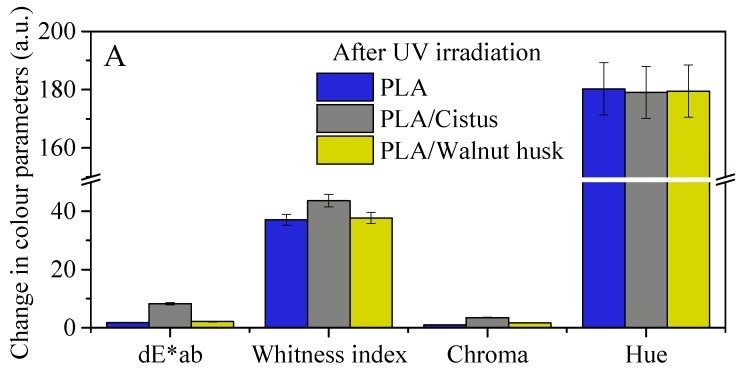
(**A**) Change of parameters of color PLA samples after UV irradiation. (**B**) Change of parameters of color PHA samples after UV irradiation.

**Table 1 polymers-11-00669-t001:** Antioxidant activity and ability to reduce transition metal ions of the extracts of *Cistus* and walnut husk.

Method		*Cistus*	Walnut Husk
**Antioxidant activity**			
**ABTS**	inhibition [%]	60.71 ± 3.04	19.84 ± 0.99
**DPPH**	inhibition [%]	57.94 ± 2.90	15.92 ± 0.80
**Reduction of transition metal ions**			
**FRAP**	Fe^3+^→Fe^2+^ [∆A, a.u.]	4.13 ± 0.21	1.85 ± 0.09
**CUPRAC**	Cu^2+^→Cu^1+^ [∆A, a.u.]	1.97 ± 0.10	0.82 ± 0.04

**Table 2 polymers-11-00669-t002:**
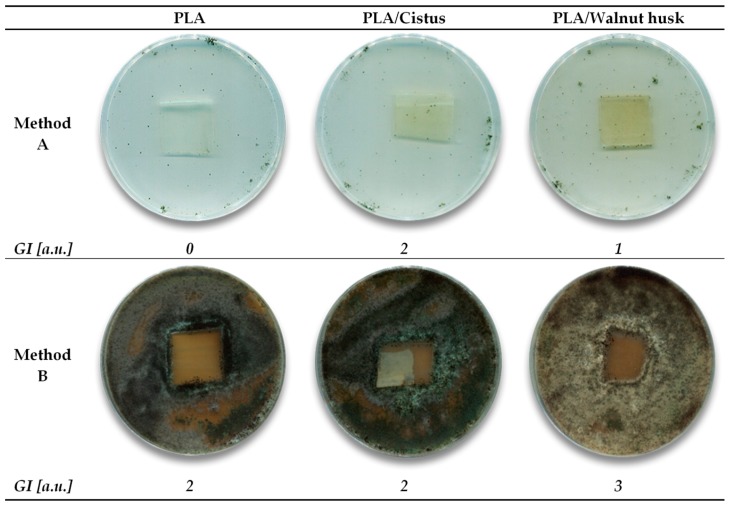
Determination of the influence of mold fungi on PLA—methods A and B (*GI—growth intensity*).

**Table 3 polymers-11-00669-t003:** Differential scanning calorimetry (DSC) analysis of PLA samples impregnated with extracts of *Cistus* and walnut husk.

Sample	*T*_g_ [°C]	Δ*H*_cc_ [J/g]	*T*_cc_ [°C]	Δ*H*_m_ [J/g]	*T*_m_ [°C]	Δ*H*_o_ [J/g]	*T*_o_ [°C]
**PLA**	58.3	23.1	107.2	25.2	146.6	15.8	**210.8**
**PLA/*Cistus***	58.7	30.4	106.5	26.7	147.3	58.2	**226.0**
**PLA/Walnut husk**	58.2	24.7	106.8	26.3	146.5	16.4	**230.7**

*T*_g_—glass transition temperature, Δ*H*_cc_—enthalpy of crystallization, *T*_cc_—crystallization temperature, Δ*H*_m_—enthalpy of melting, *T*_m_—melting temperature, Δ*H*_o_—enthalpy of oxidation, *T*_o_—initial oxidation temperature.

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
