# Peer review of "Effect of Impregnation of Biodegradable Polyesters with Polyphenols from Cistus linnaeus and Juglans regia Linnaeus Walnut Green Husk"

_polymers, 2019, doi:10.3390/polym11040669_

Round 1
Reviewer 1 Report
Authors revised the manuscripts according to the reviewers' comments. There are still some points required to be revised.
1) Authors mentioned "The impregnation efficiency can be caused by higher crystalline of the materials." Please provide the crystallinity .
2) Figure 7: why did authors take photographs at 43x magnification. Is there any small structures?
3) I understand that the reference is limited. Please write Introduction more concisely.
Author Response
Answers to Reviewer #1 comments
Reviewer #1: Authors revised the manuscripts according to the reviewers' comments. There are still some points required to be revised.:
Reviewer #1: 1) Authors mentioned "The impregnation efficiency can be caused by higher crystalline of the materials." Please provide the crystallinity.
Answer: The crystallinity of polymeric materials can be determined by differential scanning calorimetry and knowledge of the melting enthalpy of 100% crystalline 100% polymer. The degree of crystallinity Xc can be calculated as follows: Xc (%)=∆Hm/∆H0×100, where ∆Hm is the melting enthalpy of test polymer and ∆H0 is the melting enthalpy of 100% crystalline 100% PLA or PHA. The melting enthalpy estimated for 100% crystalline PLA is 97.3 J/g to 148 J/g [1]. The 148 J/g was used for calculations. The melting enthalpy of 100% crystalline PHB 146 J/g [2] was used as melting enthalpies of 100% crystalline PHA. The used PHA polymer according to the manufacturer's data is P(3,4HB). The data from the DSCs shown in our publication [3] was used to calculate the degree of crystallinity of PHA. The melting enthalpy of PHA was 40.4 J/g. The degree of crystallinity are respectively XcPHA=27% and XcPLA=17%. The results may be subject to error due to the assumed values of 100% crystallinity of polymers.
Another study confirming the more crystalline nature of PHA is WAXD analysis, however, we plan to publish these results in another publication.
[1] Farah, S.; Anderson, D.G.; Langer, R. Physical and mechanical properties of PLA, and their functions in widespread applications — A comprehensive review, Adv Drug Deliv Rev. 2016, 107, 367–92. DOI: 10.1016/J.ADDR.2016.06.012.
[2] Hu, S.; McDonald, A.G.; Coats, E.R. Characterization of polyhydroxybutyrate biosynthesized from crude glycerol waste using mixed microbial consortia. J Appl Polym Sci. 2013, 129, 1314-1321.
[3] 11. Masek, A.; Latos-Brozio, M. The Effect of Substances of Plant Origin on the Thermal and Thermo-Oxidative Ageing of Aliphatic Polyesters (PLA, PHA). Polymers. 2018, 10, 1252. DOI: 10.3390/polym10111252.
Reviewer #1: 2) Figure 7: why did authors take photographs at 43x magnification. Is there any small structures?
Answer: Photographs at 43x magnification were taken to determine if fragments of plant material are present on the surface of the samples after impregnation. Small fragments of cistus and walnut husk were not visible on the samples with the naked eye. The presence of particles of plant materials on the samples contributed to the increased biodegradability of the samples. It was unexpected result, because we expected that due to the presence of polyphenols on surface of samples fungistatic effect of polymeric materials will increase.
Reviewer #1: 3) I understand that the reference is limited. Please write Introduction more concisely.
Answer: We thank the reviewer for this suggestion. The introduction was written concisely.

Reviewer 2 Report
In this work, the authors reported a method to impregnate the degradable polymers, such as polylactide (PLA) and polyhydroxyalkanoate (PHA), with the extractions of cistus and green walnut husks. However, some fundamental problems have not been resolved; the design and performance of the manuscript are rough. In addition, the manuscript suffers from some severe limitations as follows.
1. What are the advantages of the combined extraction and impregnation process?
2. Since small pieces of plant material in the extract can promote microbial growth, it may be more reasonable to filter out excess plant debris prior to impregnation. Moreover, the adhesion of small pieces of plant material also affects the calculation of the impregnation rate.
3. Polyphenols possessed antioxidant and antibacterial properties, and PLA samples lacked a fungistatic effect after impregnation of extracts. A higher impregnation efficiency was found for the PHA compared to PLA. Did PHA samples have a fungistatic effect?
4. Polyphenols are the most important antioxidant active ingredients in extracts. Quantification of active ingredients should be carried out. More importantly, it is necessary to quantify the active ingredient in the sample after impregnation.
5. Scanning electron microscopy of the sample should be supplemented.
6. Please improve the impregnation rate and the uniformity of the impregnated sample.
7. There are still some minor shortcomings that should be corrected, such as the punctuation and space between the two words.
Author Response
Institute of Polymer and Dye Technology
Technical University of Lodz
90-924 Lodz, ul Stefanowskiego 12/16, Poland
Tel.: +48 42 631 32 23, Fax: +48 42 636 25 43
April 5, 2019
Polymers — Open Access Journal
Dear Professor,
We are resubmitting our revised paper entitled “Effect of impregnation of biodegradable polyesters with polyphenols from Cistus L. and Juglans regia L. walnut green husk” by Malgorzata Latos-Brozio and Anna Masek with a request to reconsider it for publication in "Polymers”.
We have carefully considered the Editor and Reviewers' comments. The manuscript was revised exactly according to these comments. The list of responses to the reviewer’s comments and corrections made in the manuscript is attached.
The manuscript has not been previously published, is not currently submitted for review to any other journal, and will not be submitted elsewhere before a decision is made by this journal.
For correspondence please use the following information:
corresponding author: Anna Masek
Institute of Polymer and Dye Technology
Technical University of Lodz
90-924 Lodz, ul Stefanowskiego 12/16, Poland
Tel.: +48 42 631 32 93
Fax: +48 42 636 25 43
e-mail: anna.masek@p.lodz.pl
Yours sincerely,
PhD, Dsc Anna Masek
Answers to Reviewer #2 comments
Reviewer #2: In this work, the authors reported a method to impregnate the degradable polymers, such as polylactide (PLA) and polyhydroxyalkanoate (PHA), with the extractions of cistus and green walnut husks. However, some fundamental problems have not been resolved; the design and performance of the manuscript are rough. In addition, the manuscript suffers from some severe limitations as follows.
Reviewer #2: 1. What are the advantages of the combined extraction and impregnation process?
Answer: The combination of extraction and impregnation in one process allows the introduction of active phytochemicals directly from plant material to polymers. Furthermore, the combined process of extraction and impregnation is technically much easier and cheaper than the separate extraction of plant compounds and then introducing these substances to polymers by impregnation. The combined process is performed in one apparatus, does not require special equipment and conditions. From an industrial point of view, the introduction of phytochemicals in polymers in a combined extraction and impregnation process is a much easier and more economically advantageous solution than separate processes.
Reviewer #2: 2. Since small pieces of plant material in the extract can promote microbial growth, it may be more reasonable to filter out excess plant debris prior to impregnation. Moreover, the adhesion of small pieces of plant material also affects the calculation of the impregnation rate.
Answer: We agree with the comment made by the reviewer. The small pieces of plant material in the extract can promote microbial growth and also affects the calculation of the impregnation rate. Process of filter out excess plant debris prior to impregnation is a good proposition, however, the assumption of our process was the impregnation of polymers with phytochemicals directly from plant material. Leaving the cistus and walnut husk in the impregnation process was to increase the amount of active substances in the solutions and direct adhesion of phytochemicals from plants to polymers with a carrier EtOH.
Reviewer #2: 3. Polyphenols possessed antioxidant and antibacterial properties, and PLA samples lacked a fungistatic effect after impregnation of extracts. A higher impregnation efficiency was found for the PHA compared to PLA. Did PHA samples have a fungistatic effect?
Answer: We chose PLA as samples representing all polyesters. Unfortunately, we did not do research for PHA. We agree with the reviewer. Research on PHA samples would allow comparison of both materials. However, determination of the influence of mold fungi on polymers last 3 months and we are unable to replenish them within 10 days. In addition, we do not determine the influence of mold fungi on polymers in our institute and we have financial limitations for research carried out by subcontractors. Thank you very much for the right suggestion. In the future, we will try to complete the analyzes with research suggested by the reviewer.
Reviewer #2: 4. Polyphenols are the most important antioxidant active ingredients in extracts. Quantification of active ingredients should be carried out. More importantly, it is necessary to quantify the active ingredient in the sample after impregnation.
Answer: We fully agree with the reviewer. We believe that high-performance liquid chromatography (HPLC) would be a good way to quantify individual polyphenolic compounds. Unfortunately, we intend to present detailed analyzes of plant extracts in a separate publication devoted only to analytical problems. In addition, we have financial limitations for research carried out by subcontractors.
Reviewer #2: 5. Scanning electron microscopy of the sample should be supplemented.
Answer: Due to technical and financial limitations, we are not able to perform scanning electron microscopy of the samples. Thank you very much for the right suggestion. SEM would provide valuable and important information about the surface of samples after impregnation, degree and depth of coverage of polyesters by plant extracts. In the future, we will try to complete the research description with the analysis suggested by the reviewer. At the moment we are at the stage of the initial testing of impregnation in PLA and PHA.
Reviewer #2: 6. Please improve the impregnation rate and the uniformity of the impregnated sample.
Answer: We thank the reviewer for valuable comments. As impregnation rate, we understand the impregnation time. The impregnation time was 4 hours at 50°C. Then after 4 hours, the flask and its contents were left for 24 hours at room temperature. The uniformity of the impregnated samples was evaluated on the basis of visual observations and optical microscopy. During the observation, the surface of the samples covered with plant extracts was measured and referred to the entire surface of the samples. The degree of coverage of samples with plant extracts was estimated at 90%.
Reviewer #2: 7. There are still some minor shortcomings that should be corrected, such as the punctuation and space between the two words.
Answer: Thank you for your comment. The manuscript has been read carefully and the punctuation and spaces between the two words have been corrected.
Round 2
Reviewer 2 Report
The manuscript has been well revised for several times and can be accepted in the current form.
This manuscript is a resubmission of an earlier submission. The following is a list of the peer review reports and author responses from that submission.
Round 1
Reviewer 1 Report
Authors reported the method combining the extraction of plant materials and the impregnation of polymers in one process. This suggested method should be interesting for the readers of polymers. Before the acceptance, I recommend authors to revise the manuscript according to the points as follows:
1) Authors should add more related works on the extraction and impregnation method. The major part is on the materials used in this work.
2) Could you please provide the enlarged IR spectra? It is hard to identify the peaks described in the manuscript.
3) The resolution of Figure 3 is poor.
4) From Figure 4, the impregnation efficiency was around 10%. Is this high? Authors have to compare the efficiency with the efficiency reported in the previous study.
5) Authors discussed that the difference in impregnation efficiency of PLA and PHA was due to the difference in the chemical structures of plant materials and polymers, and that the hydrophobic nature of some phytochemicals was important. Authors must provide the data or the references.
6) Authors mentioned that the antioxidant activity of Cistus was greater than that of Walnut husk, and that the impregnation efficiency was around 4 % for PLA/Cistus and 2 % for PLA/Walnut husk. Could you discuss the results of Table3 considering the antioxidant activity and the impregnation efficiency.
7) From Figure 5, authors indicated the coverage was not 100%. Authors should estimate the coverage.
8) It is hard to understand the pictures shown in Figure 8. Which part of the films are shown as the picture?
Reviewer 2 Report
In this work, the authors reported a method to impregnate degradable polymer such as polylactide (PLA) and polyhydroxyalkanoate (PHA) with the extractions of cistus and green walnut husks. The extracts contained polyphenolic compound that has an antioxidant effect. The antioxidant activity of the extracts was demonstrated by ABTS and DPPH method. After the samples were irradiated with ultraviolet (UV), the change of surface free energy and color were studied. The impregnated polymer showed thermal stability and was biodegradable, yet had no fungicidal action. Overall, the design and performance of the manuscript are common. In addition, some improvements may be necessary followed by.
1. The extracts were obtained by alcohol extraction, except for polyphenols, what were the other components? In material resistance test, what ingredients in PLA/walnut husk samples promoted the growth of microorganisms?
2. Polyphenols possessed antioxidant and antibacterial properties, and PLA samples lacked of fungistatic effect after impregnation of extracts. A higher impregnation efficiency was found for the PHA compared to PLA. Did PHA samples have fungistatic effect?
3. Phytochemicals can change the color under the influence of UV irradiation; the color parameters in Figure 6A and 6B represented the change before and after UV irradiation or just the color parameter of the sample after UV irradiation?
4. Scanning electron microscopy of the sample should be supplemented.
5. During UV irradiation, the temperature was increased to 60 ºC. Did the temperature affect the color change of the sample?
6. There are still some shortcomings should be corrected, such as the punctuation and the space between the two words. The full name should not be given again after the abbreviations appeared, such as “PLA”, “PHA” and “DSC”.
7. Some the figures are so ambiguous, it's better to improve the resolution of the figures, such as Figure 7.